# Ni Nanoparticles on the Reduced Graphene Oxide Surface Synthesized in Supercritical Isopropanol

**DOI:** 10.3390/nano13222923

**Published:** 2023-11-09

**Authors:** Yulia Ioni, Anna Popova, Sergey Maksimov, Irina Kozerozhets

**Affiliations:** 1Kurnakov Institute of General and Inorganic Chemistry, Russian Academy of Sciences, 119991 Moscow, Russia; 2Department of Chemistry, Moscow State University, 119991 Moscow, Russia

**Keywords:** reduced graphene oxide, nickel nanoparticles, supercritical fluids, nanocomposite

## Abstract

Nanocomposites based on ferromagnetic nickel nanoparticles and graphene-related materials are actively used in various practical applications such as catalysis, sensors, sorption, etc. Therefore, maintaining their dispersity and homogeneity during deposition onto the reduced graphene oxide substrate surface is of crucial importance to provide the required product characteristics. This paper demonstrates a new, reproducible method for preparing a tailored composite based on nickel nanoparticles on the reduced graphene oxide surface using supercritical isopropanol treatment. It has been shown that when a graphene oxide film with previously incorporated Ni^2+^ salt is treated with isopropanol at supercritical conditions, nickel (2+) is reduced to Ni (0), with simultaneous deoxygenation of the graphene oxide substrate. The resulting composite is a solid film exhibiting magnetic properties. XRD, FTIR, Raman, TEM, and HRTEM methods were used to study all the obtained materials. It was shown that nickel nanoparticles on the surface of the reduced graphene oxide had an average diameter of 27 nm and were gradually distributed on the surface of reduced graphene oxide sheets. The data obtained allowed us to conduct a reconnaissance discussion of the mechanism of composite fabrication in supercritical isopropanol.

## 1. Introduction

Reduced graphene oxide (RGO) is a nanomaterial obtained by the chemical or thermal reduction of graphene oxide, which is increasingly attracting the attention of scientists due to its unique properties and application prospects. The term “reduced graphene oxide” was defined in ISO/TS80004-13:2017 [1] in 2017 to distinguish it from few-layer graphene, since the material produced by the thermal or chemical reduction of graphene oxide has physical properties that are very different from those of mono- or two-layer graphene. RGO contains a large number of functional surface groups (hydroxyl, carbonyl, epoxy, and carboxyl), most often located at the edges of flakes, as well as carbon layer defects due to the specific methods of the fabrication [2,3]. Residual oxygen-containing groups, a graphene surface saturated with electron density as well as lattice defects can act as active centers for nanoparticle deposition on the RGO surface for nanocomposite synthesis [4]. Currently, such composites are becoming increasingly in-demand, since the RGO surface provides the stabilization of nanoparticles, preventing their aggregation and oxidation, and it increases their surface area, which enhances their catalytic activity [5,6,7]. At the same time, nanoparticles increase the interplanar distances between graphene flakes, preventing them from aggregating into a bulk graphite structure. Thus, the unique properties of RGO are preserved in the nanocomposite [6]. Moreover, recent studies have shown that RGO/Me composites possess properties different from those of the individual components [8,9]. This phenomenon provides a wide range of applications for composites based on RGO and metal-containing nanoparticles in sensor systems [10], for the creation of batteries and supercapacitors [11,12], in catalysis [7], and in biomedicine [13].

Metal nanoparticles on the RGO surface are capable of catalyzing a range of organic reactions [5,14,15,16]. Nickel nanoparticles (Ni NPs) are one of the most effectively used catalysts. Nickel is characterized by its relative cheapness, the ability to be extracted and reused, low toxicity, unique magnetic properties, and high activity and selectivity. However, Ni NPs with certain peculiar characteristics, as well as copper and iron NPs, are relatively difficult to synthesize, since they are often very unstable and tend to oxidize in air and form aggregates [17]. Therefore, there is a need to develop simple and reproducible techniques to obtain non-agglomerated and stable nickel nanoparticles with a well-controlled average particle size and shape gradually distributed on the surface of the RGO matrix [18]. The distribution of magnetic nanoparticles on RGO sheets potentially opens up new opportunities not only for the development of catalytic systems, but also for the creation of electrode materials [18,19] and sensors [20].

The main general methods for Ni NP composite synthesis are precipitation followed by the hydrothermal reduction of nickel chloride, acetate, or acetylacetonate in the presence of a matrix material [5,21,22]. If graphene oxide (GO) is used as a matrix, the synthesis can be carried out in one or two stages. In general, a nickel salt is deposited onto the surface of graphene oxide. After the coordination of nickel cations with graphene oxide, a reducing agent is introduced into the system, which allows us to obtain nanoparticles on the surface of GO. The GO reduction is carried out at the next stage. It is possible to combine the described stages and carry out a simultaneous reduction of the metal salt and graphene oxide. But, in this case, either strong and toxic reducing agents are used (for example, a mixture of hydrazine and sodium borohydride in an alkaline medium), or heating to high temperatures is required. Thus, in [23], it was shown that a Ni NP-decorated electrochemically reduced graphene oxide nanocomposite was fabricated on an indium-tin-oxide electrode using a facile one-pot electrochemical approach. The obtained composite showed high electro-catalytic activity when used as a sensor for glucose, exceeding the efficacy of pure components. However, there is a problem in that the average size of electrochemically obtained nickel nanoparticles has a broad size distribution within the range of 50–150 nm. Ni NPs tend to oxidize in air to form NiO, Ni_2_O_3_, Ni(OH)_2_, or NiOOH, which creates additional difficulties during synthesis and can decrease catalytic properties [24,25]. For example, 7 nm nickel nanoparticles are capable of catalyzing the Suzuki addition reaction of 4-bromoacetophenone and 4-iodoacetobenzene with 98% yield, but after the first cycle, the efficiency of this reaction drops to 1/2 (the product yield decreases to 47%) due to the oxidation process.

The use of classical wet chemistry methods, such as reduction with sodium borohydride or hydrazine, does not always lead to the formation of nickel nanoparticles. It was demonstrated in [26] that during the reduction of the GO/Ni^2+^ composite using excess NaBH_4_, nickel clusters were fixed on the RGO surface and acted as an effective catalyst in the nitrophenol reduction reaction. However, Ni NPs smaller than 2 nm typically have poor thermal stability and tend to agglomerate. Also, excessive reduction in the Ni NP diameter leads to alterations in their magnetic properties: nanoparticles switch their ferromagnetic properties to paramagnetic ones [27].

The advantage of using ferromagnetic Ni NPs on the RGO surface in heterogeneous catalysis is that the composite can be easily removed from the reaction medium for recovery and further application using a trivial permanent magnet. A similar technique was shown in [28]. In this study, RGO/Ni nanocomposite was applied to catalyze the Sonogashira cross-coupling reaction for six cycles without losing its catalytic activity. The composite was removed from the reaction mixture upon each cycle of the chemical reaction using a magnet, and then was washed and reused.

Supercritical fluids have low interfacial tension, excellent surface wetting, and high diffusion coefficients. These properties give them the possibility to be used for obtaining RGO or RGO/NPs nanocomposites [29,30]. The reduction of graphene oxide in supercritical media is an accessible, scalable, inexpensive, environmentally friendly, and safe technique. During RGO synthesis in supercritical fluids, oxygen functional groups are removed from the planar sheets of the GO, and aromatic structures are restored [31]. The mechanism of GO reduction is determined by the fact that hydrogen can be eliminated in the form of molecular H, H∙, or H^+^ at an elevated pressure and temperature. Next, dehydration can occur as a result of inter- or intramolecular reactions, the reduction of highly strained epoxide groups, decarboxylation, and the formation of a conjugated π-system.

In the present study, we report a facile and effective approach to prepare Ni nanoparticles that are gradually distributed on the surface of RGO using isopropyl alcohol under supercritical conditions. The reduction method in SCI opens a route to obtain highly dispersed crystalline nickel nanoparticles located on the surface of RGO both in the form of a powder material and in the form of an extended film.

## 2. Materials and Methods

Potassium permanganate (KMnO_4_, CAS No. 7722-64-7), Nickel (II) formate dihydrate (Ni(form)_2_·2H_2_O, CAS No. 15694-70-9), concentrated sulfuric acid (96% H_2_SO_4_, CAS No. 7664-93-9), hydrogen peroxide (30% H_2_O_2_, CAS No. 7722-84-1), and hydrochloric acid (37% HCl, CAS No. 7647-01-0) were reagent-grade (Reachem, Moscow, Russia). All reagents were used as is without purification.

### 2.1. Synthesis of GO

Graphite flakes (1 g) were dispersed in 96% sulfuric acid (60 mL) at room temperature using a mechanical stirrer. After 10 min stirring, 1 wt. eq. KMnO_4_ (1 g) was added. The mixture turned green due to the formation of the oxidizing agent MnO^3+^. Additional portions of KMnO_4_ (1 g) were added later on when the intensity of the green color of the mixture decreased, which indicated the consumption of the oxidizing agent. In total, 3.6 weight equivalent portions of KMnO_4_ were added sequentially. The end of oxidation was always determined by the disappearance of the green color after each addition of KMnO_4_. Then, about 250 mL of cold water was added to the reaction mixture, followed by the addition of 2.5 mL of hydrogen peroxide. Then, the system was washed with water and with 1M hydrochloric acid solution. The precipitate of graphene oxide was separated via a centrifuge, placed in Petri dishes, and dried in air until a constant weight was achieved.

### 2.2. Preparation of GO/Ni^2+^

The general scheme of the synthesis is shown in Figure 1a. The obtained graphene oxide (60 mg) was dispersed in 40 mL of distilled water, and then a solution of nickel diformate (90 mg Ni(form)_2_·2H_2_O in 40 mL of distilled water) was added dropwise. The resulting system was treated with ultrasound at an intensity of 1 W/cm^3^ for 10 min. Later on, water was slowly evaporated from the colloidal solution until a smooth dark brown film was obtained (Figure 1b).

### 2.3. RGO/Ni^0^ Composite Synthesis

A powder or film of the obtained GO/Ni^2+^ composite (~20 mg) was placed into a quartz test tube with 5.7 mL of 2-propanol. The test tube was placed into a steel autoclave, further heated to 280 °C, and kept for 16 h in a furnace. The temperature value of 280 °C was used to uniformly heat the thick walls of the steel autoclave, and complete conversion of graphene oxide was achieved after 16 h of heating. An increase in the pressure in the autoclave occurred simultaneously with an increase in the temperature. After cooling to room temperature, the autoclave was opened, the supernatant liquid was removed, the resulting composite was washed repeatedly with isopropyl alcohol and acetone, and then dried at a temperature of 60 °C for 5 h. The obtained composite was either a solid dark-gray film (Figure 1c), or a powder that was attracted to a permanent magnet (Figure 1d).

### 2.4. Reduction of the Nickel (II) Salt in SCI

A 100 mg sample of Ni(HCOO)_2_∙2H_2_O was placed into a quartz test tube with 5.7 mL of 2-propanol. The test tube was placed into a steel autoclave, further heated to 280 °C, and kept in the furnace for 16 h. The temperature value of 280 °C was used to uniformly heat the thick walls of the steel autoclave, and the complete conversion of graphene oxide was achieved after 16 h of heating. After cooling to room temperature, the autoclave was opened, the supernatant liquid was removed, the resulting composite was washed repeatedly with isopropyl alcohol and acetone, and then dried in air. The result was a black powder of 32 mg weight.

### 2.5. Nanomaterial Characterization

FTIR absorption spectra of the samples were recorded with a Bruker Alpha IR Fourier spectrometer with a Platinum ATR attachment in the range of 400–4000 cm^−1^ with a scanning step of 4 cm^−1^. Raman spectra were measured at room temperature with a Inspectr R532 Raman spectrometer (Moscow, Russia) combined with an Olympus CX-41 microscope (Tokyo, Japan), using a 532 nm green laser beam as the excitation source.

The identification of the phase composition of the obtained samples was carried out using a Bruker D8 Advance instrument operating in reflection mode using CuKα radiation (40 kV, 40 mA, λ = 1.54056 Ǻ) with a scanning step of 4° per minute. Transmission electron microscopy (TEM) was performed using a JEOL Jem-1011 instrument at an accelerating voltage of 80 kV. High-resolution TEM was performed using a JEOL JEM-2100F/Cs/GIF/EDS microscope at an accelerating voltage of 200 kV.

## 3. Results and Discussion

The organic solvents isopropanol and acetone in the state above the supercritical point have a unique combination of physical parameters (Table 1). The molecules differ by two hydrogen atoms, and the parameters of the supercritical state have very similar values [32,33].

Earlier works demonstrated that the combination of physical properties of the 2-propanol–acetone pair can facilitate the hydrogen transfer from isopropanol to a variety of substrates. Thus, supercritical isopropanol (SCI) can act as a donor of two hydrogen atoms, for example, in hydrogenation reactions of organic substrates. Unlike water, alcohols with large radicals are practically incapable of noticeable polymolecule cluster formation under ordinary conditions due to their lower polarity and acidity. Consequently, the system of weak hydrogen bonds in supercritical alcohols is completely destroyed, and the resulting fluid consists of alcohol molecules isolated from each other. When small amounts of substrate are introduced into the SCI medium, alcohol molecules can interact with the substrate due to thermodynamic reasons. An increased concentration of fluid molecules is achieved around each substrate molecule, and it is enclosed in a reaction “cage”. A chemical transformation occurs when molecules of the medium collide with a molecule of the substrate, like in the gas phase with effectively increased pressure. When carrying out the reaction in the SCI medium, a “concert” transition of two H atoms of the isopropanol molecule (hydrogen at the tertiary carbon and hydrogen of the hydroxyl group) to the substrate occurs. This reaction leads to the transformation of both organic and inorganic substances [34,35].

Figure 2 shows an X-ray diffraction pattern and a TEM image of the product obtained after treating nickel (II) formate in supercritical isopropanol. The diffraction pattern shows three characteristic reflections with diffraction angles 2θ = 44.66°, 51.86°, and 76.58°, which can be indexed to the (111), (200), and (220) planes corresponding to cubic crystalline Ni (JCPDS#04-0850). Thus, the process of reduction of Ni^2+^ to Ni^0^ in a supercritical environment is possible in accordance with Equation (1).
***Ni(HCOO)_2_∙2H_2_O + 2H∙ → Ni^0^ + 2HCOOH + 4H_2_O***(1)

According to the TEM image (Figure 2b), it is obvious that the formation of separated nanoparticles does not occur without the presence of a stabilizing matrix. Indeed, the fine nickel particles tend to agglomerate to a high degree even if partial stabilization by isopropyl alcohol molecules occurs.

Graphene oxide is known to be a polyfunctional ligand possessing many different oxygen-containing groups on its surface [36]. Thus, when Ni^2+^ salt is introduced into the graphene oxide water dispersion, a coordination interaction between the substances occurs. This phenomenon is manifested by the rapid formation of a GO/Ni^2+^ precipitate. Due to the large number of functional groups, the interaction of GO with Ni^2+^ occurs uniformly over the entire surface of the GO flake. A propagated composite film consisting of many GO/Ni^2+^ sheets is formed after removing the solvent.

The destruction of the GO/Ni^2+^ film does not occur after its treatment in SCI, in contrast to the process of destruction of C-O bonds during reduction with chemical reagents or temperature rise. This phenomenon is believed to be explained by the interaction of SCI with the GO molecule. In this way, two hydrogen atoms of isopropanol can deoxygenate the surface groups of GO (-OH, C=O and C-O) while compensating the formed Csp^2^-Csp^2^ bonds and reducing the number of defects of the graphene crystal lattice. Conventional heating of graphene oxide leads to the destruction of C-C bonds after the loss of oxygen groups; however, with increasing pressure, the process of formation of new C=C bonds between carbon atoms becomes more energetically favorable. At the same time, the process of reduction of Ni^2+^ ions to Ni^0^ nanoparticles occurs. The presence of nickel nanoparticles on the RGO surface imparts magnetic properties to the composite (Figure 1c,d).

Figure 3 shows the XRD data for GO, RGO, and RGO/Ni^0^. Graphene oxide has a characteristic peak in the region 2θ = 11°, which disappears after its treatment in SCI (Figure 3a,b). The peak in the region 2θ = 25° corresponds to reduced graphene oxide [37]. In the diffraction pattern of the GO/Ni^2+^ composite sample after treatment in the SCI, peaks are formed in the region 2θ = 44.74°, 52.10°, and 76.51° corresponding to Ni^0^ phase (JCPDS# 04-0850). Additionally, the GO peak disappears; instead of it, a broadened peak of RGO can be observed at 2θ = 24.40° (Figure 3c).

FTIR spectra of nickel (II) formate, GO, GO/Ni^2+^, and RGO/Ni^(0)^ composites are shown in Figure 4. The IR spectrum of the original nickel (II) formate (Figure 4a) corresponds to the literature data [38,39,40]. Graphene oxide (Figure 4b) has a characteristic broadened band in the region of 3000–3400 cm^−1^, corresponding to vibrations of -OH groups and adsorbed water molecules. In the middle of the spectrum at 1725 cm^−1^, there are stretching vibrations of C=O bonds, and the band at 1619 cm^−1^ is ascribed to the bending modes of adsorbed water molecules [41]. The fingerprint region also contains a pronounced absorption band at 1035 cm^−1^, corresponding to the stretching vibrations of the C-O bond. The spectrum of the GO/Ni^2+^ composite (Figure 4c) reveals drastic differences from the spectrum of the original GO. The disappearance of the band in the region of 1725 cm^−1^, corresponding to vibrations of the C=O bond in the carbonyl and carboxyl groups, is attributed to the coordination binding of nickel (II) cations with the GO molecule surface. The bands in the region of 1553 cm^−1^ and 1345 cm^−1^ correspond to the stretching and bending vibrations (COO^−^) of the introduced nickel (II) formate. The peak associated with vibrations of (Ni–O) is seen at 765 cm^−1^. After treatment in SCI, a spectrum typical of the fully oxygenated reduced graphene oxide is observed (Figure 4d) [42], which is in good agreement with the XRD data. The presented FTIR spectrum shows a complete absence of bands corresponding to vibrations of oxygen-containing groups of graphene oxide and vibrations of stretching and bending (COO^−^) groups in nickel (II) formate. In the region of 1400–1500 cm^−1^, a broadening of small bands is observed, characteristic of stretching vibrations of aromatic Csp^2^=Csp^2^ bonds in the graphene lattice. Thus, it is obvious that the reduction of nickel (II) formate as well as GO is complete.

As a fast, simple, and nondestructive tool, Raman spectroscopy provides valuable information on the structural properties of carbon-based materials [43]. The D band at about 1330–1350 cm^−1^ originates from the breathing modes of aroma rings activated by defects. This band is forbidden in graphite, and it becomes active in the presence of any defect in the carbon structure. The G band at 1570–1590 cm^−1^ is due to the E_2g_ phonon appearance at the Brillouin zone center. Figure 5 shows the Raman spectra of samples of the original GO, RGO obtained under supercritical conditions, and of the GO/Ni^2+^ composite before and after treatment in the SCI. All obtained spectra have characteristic D and G bands, which for the original graphene oxide correspond to 1344 cm^−1^ and 1583 cm^−1^, respectively (Figure 5a). The significant width and blue shift of the D band to 1351 cm^−1^ in the Raman spectra of GO/Ni^2+^ indicate a high concentration of structural defects caused by salt precipitation (Figure 5b). In the RGO spectrum (Figure 5c), obtained after treating graphene oxide in SCI, a decrease in the intensity of the G band and shifts of the bands to 1337 cm^−1^ and 1571 cm^−1^ are observed. The red-shift of bands in the Raman spectrum of the sample indicates the improved crystallinity of the material. In the spectrum of the RGO/Ni^0^ composite, a broadening of the D band and its slight shift to 1334 cm^−1^ are observed. The position of the G band coincides with its position in the original GO. The I_D_/I_G_ ratio in Raman spectroscopy can be used to evaluate the distance between defects in graphene materials, and for GO it is about 0.9 (under 532 nm laser excitation) and increases with the mean distance rise between two defects (Table 2). The highest I_D_/I_G_ ratio = 1.29 is found for GO after reduction with SCI. The I_D_/I_G_ ratio in the composite samples is almost completely identical and slightly exceeds it in the original graphene oxide sample (I_D_/I_G_ ratio for GO sample is about 0.90). The I_D_/I_G_ value for RGO is higher than that for GO, indicating a higher degree of in-plane defect and edge defect in the RGO due to the SCI reduction process.

In these tests, it does not matter that all samples were thin films. When interpreting the optical properties and Raman spectra of graphene-based materials, characteristics such as thickness, optical constants, band gaps, and the reflectance of both the dielectric substrate and the material under study should be taken into consideration. Due to the heterogeneity of the flake size and the distribution of oxygen-containing groups or vacancies, individual flakes of both GO and RGO are highly inhomogeneous, and, consequently, rather different values for the peak positions and widths in Raman spectra can be found in the literature [44]. It should also be noted that it is difficult to accurately compare the results in the literature because Raman spectra are often recorded from different objects (bulk, few layer or multi-layer GO and RGO), which makes it difficult to quantitatively compare the results.

Figure 6 shows TEM images of the samples of pure RGO film (Figure 6a) and the obtained RGO/Ni^0^ composite (Figure 6b). A piece of each film was dispersed in isopropanol under the ultrasound treatment. As can be seen from Figure 6b, Ni nanoparticles are firmly bound to the surface of the RGO flakes: there are no particles in a free state. Ni NPs have a crystalline structure, a completely uniform size distribution, and their average diameter is approximately 27 nm. The anisotropy of the shape of the resulting nickel nanoparticles is more clearly visible when using a higher magnification (Figure 6c,d). The HR-TEM image shows that the interplanar spacing of the lattice fringes is approximately 0.17 nm, corresponding to the (200) plane of fcc Ni. EDX analysis of the RGO/Ni^0^ composite was also performed to explore the elemental composition of the nanocomposite sample. EDX analyses at points 1 and 2 are shown in Figure 6e,f. It can be observed that RGO/Ni^0^ contained C, O, and Ni elements (insert in Figure 6e), as well as trace amounts of sulfur and nitrogen, which are part of the RGO structure (not shown in the table). According to the data obtained, the composite sample contains about 30% wt. of nickel, but this amount directly depends on the amount of nickel diformate salt introduced into the structure of graphene oxide.

Thus, it has been demonstrated that RGO obtained by SCI treatment is able to act as an effective matrix for nickel nanoparticles, which are fixed on its surface and are not released into the solvent even under ultrasound treatment.

The proposed mechanism for the synthesis of the RGO/Ni^0^ composite is shown in Figure 7. GO has many oxygen-containing surface groups. Epoxy, hydroxyl, and carbonyl groups are located over the entire surface of GO, while carboxyl groups are located at the edges of the plane and internal defects. Nickel (II) formate is an organic salt with a Ni^2+^ cation and two COOH^−^ anions (Figure 7a). After merging the dispersions, ligand exchange occurs. The carboxyl groups of GO (or a combination of carbonyl and hydroxyl groups located closely) are able to replace one or two formate anions, resulting in coordination with the nickel (II) cation (Figure 7b). The hydrogen atom originating from isopropanol after reaching the supercritical point is able to hydrogenate the C-O and O-H bonds in the GO/Ni^2+^ composite. After the elimination of by-products (water and carbon dioxide), the formation of C-C bonds occurs in the reduced graphene oxide. In this way, new defects can be formed, for example, five-membered rings (Figure 7c). This process explains the increased I_D_/I_G_ ratio in the Raman spectrum of reduced graphene oxide. At the same time, the reduction of Ni^2+^ to Ni^0^ also occurs under the influence of elevated temperature and pressure. The resulting nickel atoms and clusters become centers of nucleation, and the subsequent growth of nanoparticles occurs on the RGO surface.

To confirm the effect of SCI treatment, the GO/Ni^2+^ composite was reduced by simply heating a film sample to 280 °C. TEM images are shown in Appendix A section. Thermal reduction of the film in the absence of supercritical conditions does not lead to the formation of uniformly distributed Ni nanoparticles on the surface of the RGO. Most of the RGO flake surface is free of nanoparticles or contains dark agglomerates (Appendix A). Some small areas up to 300 nm are partially covered with nickel nanoparticles (Appendix A), formed after the thermal decomposition of nickel (II) formate. However, such areas are not regular, and the resulting sample does not respond to the applied magnetic field, as was the case for samples after treatment in a supercritical fluid.

## 4. Conclusions

As a result of this study, a new facile method for obtaining small Ni^0^ nanoparticles, with controlled size and uniform distribution, on a reduced graphene oxide surface by using supercritical isopropanol both as a medium and a reducing agent was demonstrated. In the process of increasing temperature and pressure up to the supercritical point, acetone is formed with the release of hydrogen. This leads to the simultaneous reduction of nickel (II) formate and the deoxygenation of oxygen-containing functional groups of graphene oxide. Also, when carrying out the process in the SCI medium, the surface defects of graphene oxide are healed, which opens possibilities to obtain a reduced graphene oxide material with larger lateral flakes. The surface of reduced graphene oxide sheets acts as a matrix for nickel nanoparticles and prevents their further growth and agglomeration. The reduction of graphene oxide in supercritical isopropanol media allows the reaction to be carried out at a relatively low temperature (280 °C) with 100% conversion. In addition, when using supercritical fluids, there is no formation of toxic by-products that can be obtained when using, for example, sodium borohydride. Thus, reduction in supercritical isopropyl alcohol is an inexpensive, scalable, and environmentally friendly technology for producing composites based on nickel nanoparticles and reduced graphene oxide. The resulting composite material with potentially good conductive and magnetic properties can be used as a catalyst for hydrogenation reactions of unsaturated hydrocarbons and also as a sorbent of organic toxic substances in filtration systems.

## Figures and Tables

**Figure 1 nanomaterials-13-02923-f001:**
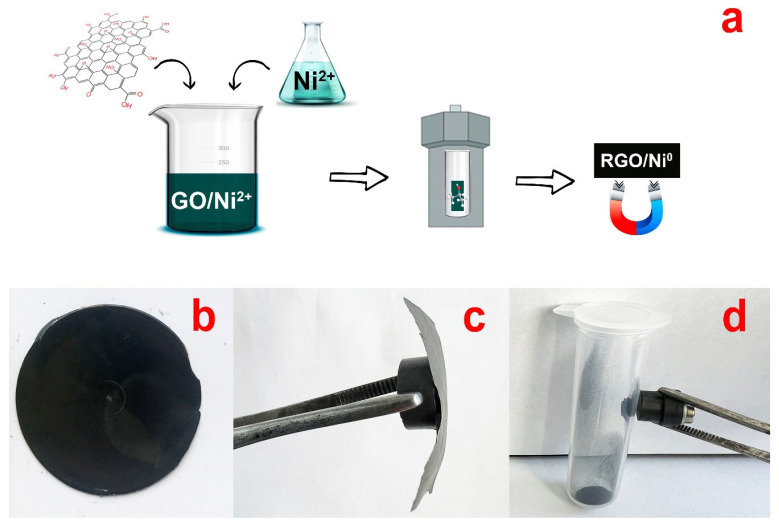
General scheme for the synthesis of RGO/Ni^0^ composite (**a**) and photos of the GO/Ni^2+^ film (**b**), and the RGO/Ni^0^ film (**c**) and powder (**d**) held by the permanent magnet.

**Figure 2 nanomaterials-13-02923-f002:**
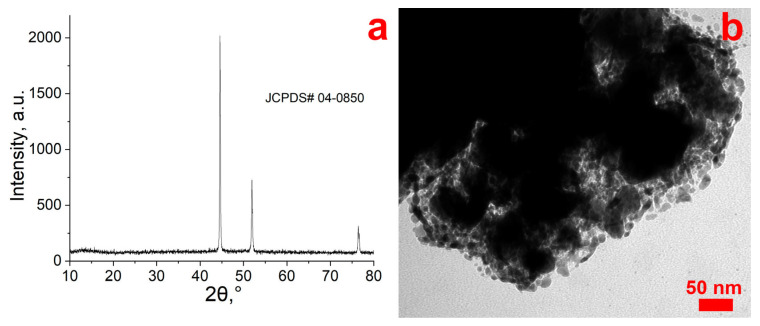
XRD (**a**) and TEM (**b**) of Ni (II) formate salt after SCI treatment.

**Figure 3 nanomaterials-13-02923-f003:**
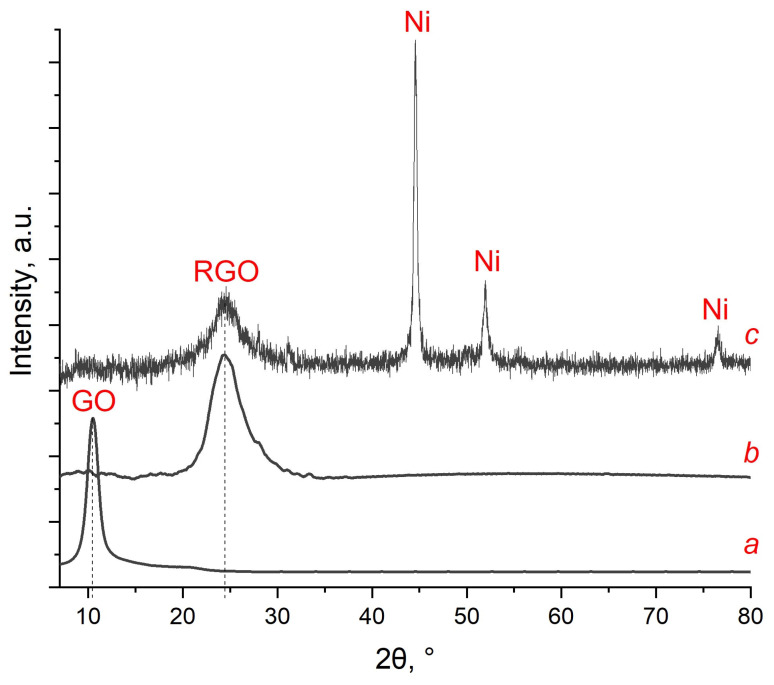
XRD of GO (**a**), RGO (**b**), and RGO/Ni^0^ (**c**).

**Figure 4 nanomaterials-13-02923-f004:**
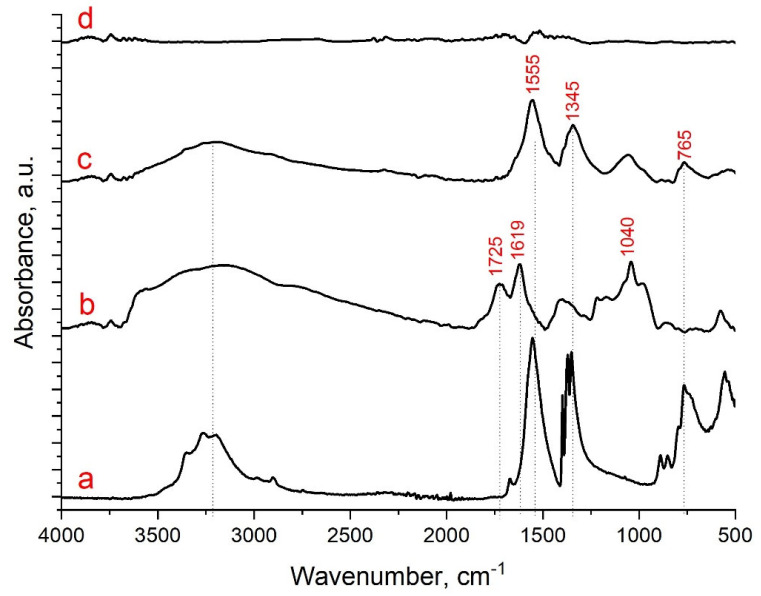
FTIR of Ni (II) formate (**a**), GO **(b**), GO/Ni^2+^ (**c**), and RGO/Ni^0^ (**d**).

**Figure 5 nanomaterials-13-02923-f005:**
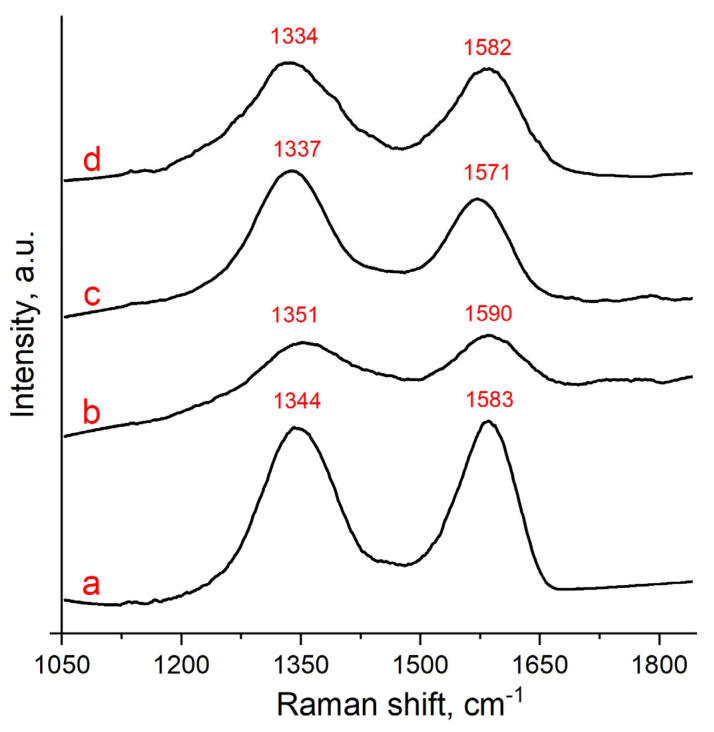
Raman spectra of GO (**a**), GO/Ni^2+^ (**b**), RGO (**c**), and RGO/Ni^0^ (**d**).

**Figure 6 nanomaterials-13-02923-f006:**
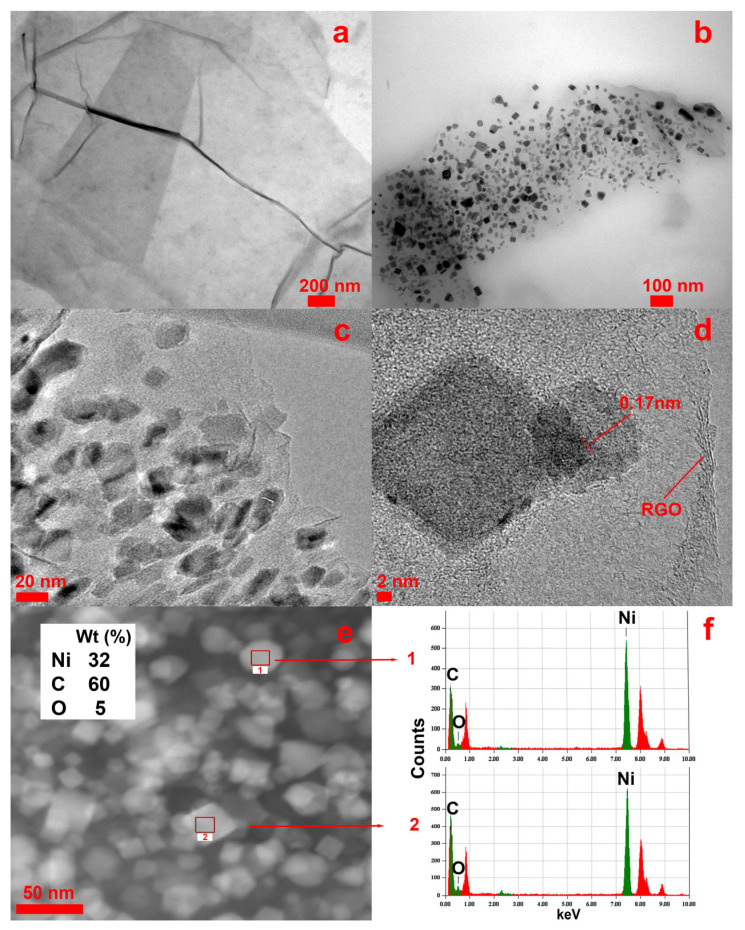
TEM of RGO (**a**) and RGO/Ni^0^ (**b**), HRTEM of RGO/Ni^0^ (**c**–**e**), and EDX of plots 1 and 2 of RGO/Ni^0^ sample (**f**).

**Figure 7 nanomaterials-13-02923-f007:**
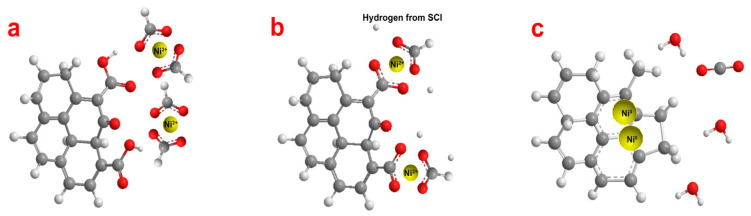
Mechanism for the synthesis of the RGO/Ni^0^ composite. GO and nickel diformate structure (**a**), the ligands change process (**b**), reduction of Ni (2+) and NP nucleation on the RGO surface (**c**). Gray, white, and red balls are carbon, hydrogen, and oxygen atoms, respectively.

**Table 1 nanomaterials-13-02923-t001:** Supercritical parameters of isopropanol and acetone.

	*Iso*-Propyl Alcohol (C_3_H_8_O)	Acetone (C_3_H_6_O)
T_c_, °C	235.3	235.0
P_c_, bar	47.6	47.0
D_c_, g/mL	0.273	0.279

T_c_, P_c_ and D_c_—temperature, pressure, and diffusion coefficient after critical point, respectively.

**Table 2 nanomaterials-13-02923-t002:** Ratio of intensities of D and G bands for the samples.

	GO	GO/Ni^2+^	RGO	RGO/Ni^0^
I_D_/I_G_	0.90	1.03	1.29	1.02

## Data Availability

The data presented in this study are available on request from the corresponding author.

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
