# Peer review of "Ni Nanoparticles on the Reduced Graphene Oxide Surface Synthesized in Supercritical Isopropanol"

_nanomaterials, 2023, doi:10.3390/nano13222923_

Round 1

Reviewer 1 Report

Comments and Suggestions for Authors

The topic of this manuscript is interesting. However, there are some issues still need to be solved and major revisions are required.

1.     Graphene oxide could be reduced by various reductants and methods. What are the advantages of supercritical isopropanol compared to other methods?

2.     The language needs to be further polished. For example, “nickel (II) formate is reduced to Ni0” in line 15 should be revised as “nickel2+ is reduced to Ni0”.

3.     The introduction section is a little long. Please revise the introduction section into less than 3 pages.

4.     RGO and its composites are widely applied in many fields. Some recently published references are suggested to be cited to enrich the content, e.g. Frontiers in Chemistry 2020, 8, 603; Molecules 2020, 25 (12); Colloids and Surfaces A: Physicochemical and Engineering Aspects 2020, 602.

5.     The purity of the reagents should be offered in “2. Materials and Methods”.

6.     How about Ni salt content on the property of resulting Ni/RGO?

7.     Why does supercritical isopropanol process fix at 280°C and kept for 16 hours? How about the influence of other temperatures?

8.     Please pay attention to the writing of superscript and subscript.

9.     It is strange that the ID/IG of RGO is larger than that of GO. Please double check the data.

10.  What are the potential applications of as prepared Ni/RGO? Please give some prospects in the introduction or conclusion section.

11.  Many of the references are too old and not closely related to the theme of this manuscript. Please delete some of them.

Comments on the Quality of English Language

Moderate editing of English language is required.

Author Response

Dear Reviewer,

thank you for your time and your interest in our work. Your fruitful comments allow us to improve the quality of the manuscript.

To responses to your comments are listed below

The topic of this manuscript is interesting. However, there are some issues still need to be solved and major revisions are required.

  1. Graphene oxide could be reduced by various reductants and methods. What are the advantages of supercritical isopropanol compared to other methods?

Answer: The reduction of graphene oxide using supercritical isopropanol allows the reaction to be carried out at a relatively low temperature (only 280°C) with 100% conversion. In addition, when using supercritical fluids, there is no formation of toxic by-products that can be obtained when using, for example, sodium borohydride. Thus, SCI reduction is an inexpensive, scalable and environmentally friendly technology for producing composites based on nickel nanoparticles and reduced graphene oxide.

Corresponding changes have been incorporated into the text.

  1. The language needs to be further polished. For example, “nickel (II) formate is reduced to Ni0” in line 15 should be revised as “nickel2+ is reduced to Ni0”.

Answer: Thank you, the text has been edited and appropriate corrections have been made.

  1. The introduction section is a little long. Please revise the introduction section into less than 3 pages.

Answer: Thank you, the Introduction has been reduced to 3 pages.

  1. RGO and its composites are widely applied in many fields. Some recently published references are suggested to be cited to enrich the content, e.g. Frontiers in Chemistry 2020, 8, 603; Molecules 2020, 25 (12); Colloids and Surfaces A: Physicochemical and Engineering Aspects 2020, 602.

Answer: Thank you very much for your comment, references have been added to the text

  1. The purity of the reagents should be offered in “2. Materials and Methods”.

Answer: Thank you for your note, reagent purity has been added to the Materials and methods section

  1. How about Ni salt content on the property of resulting Ni/RGO?

Answer: Nickel content is added in Fig. 6e and in the text of the manuscript.

  1. Why does supercritical isopropanol process fix at 280°C and kept for 16 hours? How about the influence of other temperatures?

Answer: The supercritical point for isopropanol is 235.3ºÐ¡, however, exposure at 280ºÐ¡ was used in order to uniformly heat the thick walls of the autoclave. 100% conversion of graphene oxide reduction was achieved in 16 hours; it is not economically feasible to use a longer holding time. Appropriate explanations have been included in the text.

  1. Please pay attention to the writing of superscript and subscript.

Answer: Thanks for the note, corrections have been made

  1. It is strange that the ID/IG of RGO is larger than that of GO. Please double check the data.

Answer: Thank you very much for your comment. Indeed, reduction in a supercritical environment leads to strong disordering of layers and the formation of edge defects, which is expressed in the ratio ID/IG = 1.29. In addition, studies were carried out for the resulting films with a thickness of 100 μm. Therefore, the data presented are the obtained experimental characteristics of new materials. Explanations are included in the text.

  1. What are the potential applications of as prepared Ni/RGO? Please give some prospects in the introduction or conclusion section.

Answer: Thanks for your comment, alterations have been incorporated into the text.

  1. Many of the references are too old and not closely related to the theme of this manuscript. Please delete some of them.

Answer: Thank you very much for your comments, the references have been corrected

Reviewer 2 Report

Comments and Suggestions for Authors

The experimental studies need to be more systematic and scientifically should be provided in order to claim for strong enough to support so that, this article reader will get attract more attention which are lacking this current form of this manuscript, especially

1. The current manuscript doesn’t provide in the new application studies, the application studies need to be scientifically justified with more experimental studies, which are lacking this current form of this manuscript, I suggest author to has to bring new insightful Ni nanoparticles on the reduced graphene oxide concept its quite well established one. What is advantage of this current study, please provide

2. In the characterization part in order to prove the magnetic properties i suggest author has to include the following details are please provide Superconducting Quantum Interference Device, (SQUID) of each material of NPs (as alone Ni nanoparticles) before and after modification (Ni nanoparticles on the reduced graphene oxide) compare your results and give your detailed explanations and also in Figure 4d, RGO/Ni0 FT-IR spectra, why most of the peaks were disappeared? this need to be justified?

In Figure 6f, RGO/Ni0 Please provide elemental composition along with EDX spectra and justified your answers

3. I suggest author has to provide schematic representation of mechanism of interaction, and also author has to showcase their conceptual illustration showing their interaction the following article will be useful for authors information

4. please provide your experimental advantage and compare your data’s with previous studies, and itemize advantage of your current study.

5. The reduced graphene oxide become more hydrophobic behaviour how do the achieved their ongoing studies? how this reduced graphene oxide would be helpful for author has to be justify/prove their explanations?

6. Most of references are not relevant to the current content of manuscript especially from item from 1 to 4, 16 its just exhaust rated up to 63 references, the graphene oxide materials its well established one what’s your new insightful which you would like to address which are missing

7. The conclusion and introduction need to be re-written and revise, after incorporating the above mentioned comments I suggest the editor can consider further.

Author Response

Dear Reviewer,

thank you for your time and your interest in our work. Your fruitful comments allow us to improve the quality of the manuscript.

To responses to your comments are listed below

The experimental studies need to be more systematic and scientifically should be provided in order to claim for strong enough to support so that, this article reader will get attract more attention which are lacking this current form of this manuscript, especially

  1. The current manuscript doesn’t provide in the new application studies, the application studies need to be scientifically justified with more experimental studies, which are lacking this current form of this manuscript, I suggest author to has to bring new insightful Ni nanoparticles on the reduced graphene oxide concept its quite well established one. What is advantage of this current study, please provide.
  2. please provide your experimental advantage and compare your data’s with previous studies, and itemize advantage of your current study.

Answer: The reduction of graphene oxide using supercritical isopropanol allows the reaction to be carried out at a relatively low temperature - only 280°C with 100% conversion. In addition, when using supercritical fluids, there is no formation of toxic by-products that can be obtained when using, for example, sodium borohydride. Thus, SCI reduction is an inexpensive, scalable and environmentally friendly technology for producing composites based on nickel nanoparticles and reduced graphene oxide.

Corresponding changes have been incorporated into the text.

  1. In the characterization part in order to prove the magnetic properties i suggest author has to include the following details are please provide Superconducting Quantum Interference Device, (SQUID) of each material of NPs (as alone Ni nanoparticles) before and after modification (Ni nanoparticles on the reduced graphene oxide) compare your results and give your detailed explanations and also in Figure 4d, RGO/Ni0 FT-IR spectra, why most of the peaks were disappeared? this need to be justified?

Answer: Thank you very much for your comment. Unfortunately, due to the short response time to the review, it is impossible to provide the results of this study. But this valuable remark will definitely be taken into account in future publications.

After SCI treatment the GO/Ni2+ composite, the surface of graphene oxide is reduced, which is expressed in the disappearance of stretching and deformation vibrations corresponding to hydroxyl, carbonyl and epoxy groups. The reduction of Ni2+ to Ni0 also leads to the disappearance of carboxyl groups in the region of 1400-1550 cm-1, as well as vibrations of Ni-O-C bonds in the FTIR spectrum of the RGO/Ni0 composite. Thus, the final spectrum of the RGO/Ni0 composite is an almost straight line with weak peaks from vibrations of aromatic C=C bonds in the region of 1500 cm-1. The corresponding text has been added to the manuscript.

In Figure 6f, RGO/Ni0 Please provide elemental composition along with EDX spectra and justified your answers

Answer: Thank you, the contents of the elements have been added to the text and to the figure. 6

  1. I suggest author has to provide schematic representation of mechanism of interaction, and also author has to showcase their conceptual illustration showing their interaction the following article will be useful for authors information

Answer: thank you very much for your comment. The illustration (fig. 7) has been added to the work.

Corresponding changes have been incorporated into the text.

  1. The reduced graphene oxide become more hydrophobic behaviour how do the achieved their ongoing studies? how this reduced graphene oxide would be helpful for author has to be justify/prove their explanations?

Answer: RGO is hydrophobic, thus expanding the range of use to carry out catalytic reactions in non-polar environments

  1. Most of references are not relevant to the current content of manuscript especially from item from 1 to 4, 16 its just exhaust rated up to 63 references, the graphene oxide materials its well established one what’s your new insightful which you would like to address which are missing

Answer: Thank you very much for your comments, the references have been corrected

  1. The conclusion and introduction need to be re-written and revise, after incorporating the above mentioned comments I suggest the editor can consider further.

Answer: Thank you very much, the necessary changes have been made to the text

Round 2

Reviewer 1 Report

Comments and Suggestions for Authors

The manuscript could be accepted now.

Reviewer 2 Report

Comments and Suggestions for Authors

Reviewer comments

The reviewer thanks to the authors for this revised version that address most of the comments made in its previous report. The manuscript can be published as it stands.